# Mechanisms of EGFR Resistance in Glioblastoma

**DOI:** 10.3390/ijms21228471

**Published:** 2020-11-11

**Authors:** Peter C. Pan, Rajiv S. Magge

**Affiliations:** 1Division of Neuro-Oncology, NewYork-Presbyterian/Columbia University Irving Medical Center, New York, NY 10032, USA; 2Division of Neuro-Oncology, NewYork-Presbyterian/Weill Cornell Medicine, New York, NY 10021, USA; ram9116@med.cornell.edu

**Keywords:** epidermal growth factor receptor, treatment resistance, heterogeneity, glioblastoma

## Abstract

Glioblastoma (GBM) is the most common primary malignant brain tumor in adults. Despite numerous efforts to target epidermal growth factor receptor (EGFR), commonly dysregulated in GBM, approaches directed against EGFR have not achieved the same degree of success as seen in other tumor types, particularly as compared to non-small cell lung cancer (NSCLC). EGFR alterations in glioblastoma lie primarily in the extracellular domain, unlike the kinase domain alterations seen in NSCLC. Small molecule inhibitors are difficult to develop for the extracellular domain. Monoclonal antibodies can be developed to target the extracellular domain but must contend with the blood brain barrier (BBB). We review the role of EGFR in GBM, the history of trialed treatments, and the potential paths forward to target the pathway that may have greater success.

## 1. Introduction

Glioblastoma (GBM) is an aggressive primary brain tumor. There are over 10,000 new diagnoses of GBM in the United States each year. Less than 20% of patients survive to two years [1]. The modern treatment paradigm consists of maximal safe surgical resection followed by fractionated radiotherapy with concurrent daily temozolomide, and subsequent adjuvant temozolomide given on a 5 day on and 23 day off schedule for 6–12 cycles [2]. Approach to treatment of GBM has changed little in the intervening years, despite hundreds of interventional trials since this paradigm was first published in 2005. Further temozolomide does not significantly improve outcomes in patients whose tumors do not demonstrate MGMT (enzyme O6-methylguanine-DNA methyltransferase) promoter methylation [3,4,5,6].

Concerted efforts at targeting specific genetic alterations in glioblastoma have largely failed. Chief among these targets is epidermal growth factor receptor (EGFR), the most commonly altered receptor tyrosine kinase and one of the most common alterations in glioblastoma [7]. Attempts to target EGFR in glioblastoma have had minimal success, especially in comparison to other tumors such as non-small cell lung cancer (NSCLC). The failure of EGFR-directed treatments and other targeted therapies has been ascribed to the high intra-tumoral heterogeneity of glioblastoma, but other obstacles likely play a role as well.

This review focuses on both the history and future of targeting EGFR in glioblastoma, with a special emphasis on tyrosine kinase inhibitors that have already achieved success in EGFR mutant NSCLC.

## 2. Results

### 2.1. Epidermal Growth Factor Receptor (EGFR) Structure

Epidermal growth factor receptor is an 1186-residue transmembrane protein, 170 kDa in mass [8]. It is a member of the ErbB (erb-b2 receptor tyrosine kinase) or HER (human epidermal growth factor receptor) family. This family of receptors consists of four members—EGFR (ErbB1 or HER1), ErbB2 (Neu or HER2), ErbB3 (HER3), and ErbB4 (HER4) [9]. EGFR starts as a 1210-residue precursor that is processed to the mature 1186-residue protein, after cleavage of the N-terminal sequence. N-linked glycosylation is then required for translocation to the cell surface [8]. EGFR exists as an inactive monomer, and classically undergoes ligand-induced dimerization, kinase domain activation, and downstream tyrosine phosphorylation to affect the inner state of the cell. Dimerization can occur as either homodimers or heterodimers with other receptors in the family [8].

The extracellular portion of the receptor consists of four domains tethered in a closed configuration. Ligand binding results in allosteric changes that untether the domains and reveal the dimerization arm on domain II. This arm interacts with a pocket at the base of the partner receptor’s domain II loop as well as with domains I and III (L1 and L2) of the partner receptor, forming the active dimer [8,10,11,12,13]. A homodimer of two receptors in the open configuration is illustrated [10,14] (Figure 1).

On the intracellular domain, the kinase binding pocket where ATP (adenosine triphosphate) binds is occluded by an activation loop. Once activated, the regulatory C-helix—also known as αC-helix—switches from an “out” position to an “in” position. The activation loop also moves to expose the binding pocket. These changes form the active kinase configuration [14,15] (Figure 2). In the same transition to the active state, a critical and highly-conserved triplet Asp-Phe-Gly, known as DFG, which is normally in the out position (DFG-out), rotates inward into position where it can complex with magnesium and coordinate phosphate transfer from ATP (DFG-in). Small molecule tyrosine kinase inhibitors function by occupying the kinase binding pocket, preventing ATP binding. Inhibitors can have higher affinity for one kinase conformation or the other—for example, erlotinib and gefitinib have higher affinity for the active conformation (with αC-helix in), while lapatinib has higher affinity for the inactive conformation (with αC-helix out) [16]. In the context of EGFR kinase domain inhibitors, the former are generally referred to as type I inhibitors, and the latter as type II inhibitors [17] (though it should be noted that nomenclature is variable, with lapatinib being considered a type 1 ½ inhibitor by Roskoski’s classification [18], because with lapatinib the DFG-motif is still inward even though the αC-helix is displaced outward).

Ligand binding on the extracellular surface is thought to appropriately orient domains I, II, and III (L1, CR1, and L2) in such a way that positions domain IV (CR2) and the tyrosine kinase domain for activation [6]. The carboxy lobe interacts with the amino lobe of its partner and induces conformational changes, with helix αH on the activating side interacting directly with αC (C-helix) on the partner, rotating αC to the activated inward position [7].

EGFR has numerous ligands including epidermal growth factor (EGF), transforming growth factor alpha (TGF-α), and heparin-binding EGF-like growth factor (HB-EGF) (Figure 3). ErbB3 and ErbB4 bind neuregulins, also known as heregulins or neu differentiation factors [19]. The differences in the C-terminal domain drive differences in ligand specificity [8]. Once activated, signal strength and duration are controlled by receptor internalization and recycling [20].

EGFR and its partners phosphorylate a range of downstream effectors [21,22,23] (Figure 4). Examples include the MAPK (mitogen-activated protein kinase) pathway by way of the adaptor protein Grb2 (growth factor receptor-bound protein 2), the JAK (Janus kinase) and STAT (signal transducer and activator of transcription) pathway, PLCγ (phospholipase Cγ) and PKC (protein kinase C), and the PI3K (phosphoinositol-3 kinase) and Akt (or PKB, protein kinase B) pathway [8,24]. The effective number of downstream targets is expanded by heterodimerization of EGFR with other members of the ErbB family.

EGFRvIII, deletion of exons 2 through to 7 (residues 6–273), is the most common EGFR mutation in glioblastoma. Deletion of exons 2 through 7 leaves a constitutively active EGFR, driving growth in a cell-autonomous manner. It is suggested that the receptor sequence needed for endocytosis is lost in the EGFRvIII variant [25]. The EGFRvIII mutant is classically thought of as being low constitutively active [25], however there is data to suggest that EGFRvIII dimers are as active as wild-type ligand-bound activated EGFR [26]. As deletion of exons 2 through 7 results in constitutive activation, domain I and domain II are thought to be responsible for inhibition of kinase activation. EGFRvIII and other ectodomain alterations appear to lead to preferential adoption of an intermediate conformation (between open and closed) of the ectodomain [27]. This intermediate ectodomain configuration is the target of ABT-414 (through the targeting antibody ABT-806, which binds to the exposed epitope on this intermediate state).

### 2.2. Targeting EGFR

EGFR mutations in NSCLC are characterized primarily by alterations in the intracellular tyrosine kinase domain. Treatments directed at these mutations represent some of the earliest successes in EGFR targeting. Specific alterations include mutations in exon 21 (the L858R point mutation) and in-frame deletions in exon 19. L858R mutation destabilizes the inactive conformation and stabilizes the usually disordered αC-helix of the wild-type EGFR monomer at the dimerization interface. This alters the free energy landscape in favor of dimer formation [28,29,30] but comes at the cost of reduced affinity of the L858R mutant EGFR kinase for ATP [29]. First-generation 4-anilinoquinazolines TKIs (tyrosine kinase inhibitors) such as erlotinib and gefitinib capitalize on this weakness by acting as ATP-competitive reversible inhibitors at the kinase domain and outcompeting ATP for the mutant L858R EGFR—resulting in inhibition of downstream EGFR signal transduction [16,31].

After an initial period of response to the first-generation kinase inhibitors, tumors acquire secondary resistance mutations [32,33,34]. T790M accounts for about half of these resistance mutations [35,36]. This was initially thought to act as a steric gatekeeper to inhibitor binding [37], but it was later shown that gefitinib maintained similar affinity to the kinase domain regardless of whether T790M was present—suggesting that T790M does not sterically hinder inhibitor binding. Instead, it was shown in the same paper that T790M markedly increases and restores the affinity of L858R for ATP [38]. First-generation inhibitors are unable to outcompete ATP for the L858R-T790M mutant and are therefore ineffective.

Afatinib, the first clinically-approved 2nd generation inhibitor, is a 4-anilino-quinazoline-based irreversible inhibitor designed to covalently modify the ATP-binding site of the kinase domain EGFR irreversibly at Cys 797—rendering the target inactive [39,40]. Dacomitinib is another 2nd generation inhibitor that covalently binds to Cys 797 like afatinib. Although 2nd generation inhibitors showed excellent potency against L858R-T790M double mutants, side effects due to equally-potent inhibition of the wild-type EGFR (found in normal tissues) were ultimately dose-limiting with rash, diarrhea, mucositis, and ocular toxicities being among the most common [41,42].

Third generation inhibitors were developed to target T790M and spare wild-type EGFR. Osimertinib, a 3rd generation inhibitor selected for affinity to T790M and to the exclusion of the wild-type EGFR kinase, successfully minimized off-target effects and greatly improved the pharmacokinetic and pharmacodynamic profile [43]. Excellent blood brain barrier penetration, superior to that of other available inhibitors, has earned osimertinib a role in managing brain metastases [43]. Rociletinib is another 3rd generation inhibitor similar to osimertinib, but with additional action against insulin growth factor receptor 1 (IGFR1) [44].

Resistance mutations to 3rd generation inhibitors include C797S point mutation (mutating the exact site of covalent bond formation required by irreversible inhibitors) and activation of escape pathways in MET (Met receptor tyrosine kinase), KRAS (Kirsten rat sarcoma), MAPK, PI3K (phosphoinositide-3 kinase), CDK (cyclin-dependent kinase) 4/6, or RET (rearranged during transfection receptor) [45]. Next-line EGFFR inhibitors are in development. EAI045 is a so-called 4th generation inhibitor in development with reported efficacy against C797S—although this is pending further confirmatory studies, as it does not appear to have been specifically designed to target the C797S variant, and only demonstrated efficacy when co-administered with cetuximab [46]. As inhibitors become more selective and carry fewer side effects, combination treatments become possible—for example, combined targeting of EGFR and MET (with combination osimertinib and crizotinib, or osimertinib and savolitinib [47,48,49]) or combined targeting of EGFR and RET [50]. Furthermore, other EGFR alterations, such as exon 20 insertions, have always presented a therapeutic challenge [51], and continue to represent an area of active therapeutic investigation.

Unfortunately, similar successes have not been seen in glioblastoma. The experience of EGFR targeting in glioblastoma is presented in detail below, and their mechanisms of action are summarized (Table 1). A list of EGFR trials in GBM is also provided (Table 2).

### 2.3. Specific EGFR Targeted Agents

#### 2.3.1. Gefitinib

Gefitinib is a 1st generation reversible ATP-site competitive EGFR kinase inhibitor.

Mayo/NCCTG (North Central Cancer Treatment Group) N0074 was a phase II study assessing 98 patients with newly-diagnosed glioblastoma [60]. Patients were treated with monotherapy gefitinib at 500 mg daily, escalated to 1000 mg daily if on steroids or on CYP (cytochrome P450) 3A4-inducing agents. Twelve-month progression free survival (PFS) was 16.7%, and 12-month overall survival (OS) was 54.2% on treatment, similar to historical controls. EGFR status (amplification or EGFRvIII) was not associated with outcome.

RTOG (Radiation Therapy Oncology Group) 0211 was a phase I/II study investigating the combination of radiotherapy and concurrent gefitinib (dosed at 500 mg daily, up to 750 mg daily if on enzyme-inducing anti-epileptic drugs) in patients with newly-diagnosed glioblastoma [61]. No overall survival benefit was found compared to historical outcomes of radiotherapy alone. Pre-treatment EGFR expression did not correlate with outcome.

An open-label phase II single-center study of 53 patients with recurrent glioblastoma (not selected by EGFR status) found no objective tumor response with gefitinib at 500 mg daily (with protocol to escalate to 750 mg then 1000 mg as tolerated), though it was well-tolerated [62]. Epidermal growth factor expression again did not correlate with outcome.

Post-treatment tumor samples were not evaluated for appropriate inhibition of EGFR in either of the above studies.

#### 2.3.2. Erlotinib

Similar to gefitinib, erlotinib is a first-generation reversible ATP-site competitive EGFR kinase inhibitor.

A phase II open label study at the University of California San Francisco, published in 2009, evaluated 65 GBM patients treated with erlotinib and temozolomide (TMZ) during and after radiotherapy [63]. Erlotinib was dosed at 100 mg daily during radiotherapy, and 150 mg daily post-radiotherapy (and escalated until a maximum tolerated dose 200 mg daily). For the 28 patients not on enzyme-inducing antiepileptic drugs (EIAED), 22 patients were treated with adjuvant erlotinib to at least 150 mg daily. For the 37 patients on EIAEDs, 27 patients were treated to 300 mg daily or less; the other 10 patients were treated at 350–400 mg daily. Median overall survival was 19.3 months (versus historical 14.1 months). Pre-treatment EGFR did not correlate with survival (non-statistically significant hazard ratio of 0.65 in MGMT unmethylated patients was noted).

N0177 was a phase I/II multicenter study evaluating erlotinib with temozolomide and radiotherapy in 81 newly-diagnosed glioblastomas, published in 2008 [64]. Patients on EIAEDs were excluded. Erlotinib was dosed at 150 mg daily continuously through radiotherapy and adjuvant treatment. Of the 81 patients, 30 were noted to have the EGFRvIII variant. Two patients were missing EGFR FISH (fluorescence in situ hybridization), 7 had normal EGFR FISH, and the remainder (the majority, at 72 of 81 patients) had gain of chromosome 7, amplification of EGFR, or duplicate EGFR—with gain of chromosome 7 noted to be the most common. Overall survival at 12 months was 61%, comparable to EORTC (European Organization for Research and Treatment of Cancer) 26,981 [2]. Pre-treatment EGFR did not correlate with outcome.

A Cleveland Clinic single-center study (published in 2010) treated 27 patients with newly-diagnosed GBM with concurrent radiotherapy and temozolomide followed by erlotinib at 50 mg daily (and up to 150 mg daily) [52]. Patients on EIAEDs were excluded. The median progression free survival was 2.8 months. An unacceptably high death rate prompted early closure of the study, however the authors commented that the addition of erlotinib did not seem to be the cause of increased rate of death in the trial.

A 2009 multicenter study of erlotinib in recurrent higher grade gliomas (glioblastoma, anaplastic astrocytoma, anaplastic oligodendroglioma) and stable glioblastomas post-radiotherapy [65]. Patients on EIAEDs were excluded. Patients were treated to erlotinib 150 mg daily. Fifty-three patients were in the recurrent group, and 43 patients were in the stable group. Progression free survival at 6 months in the recurrent glioblastoma population was 3%. Outcome did not correlate with pre-treatment EGFR.

A phase II study of erlotinib 150 mg daily in 13 recurrent glioblastoma patients that were PTEN positive, EGFR positive, and EGFRvIII positive by immunohistochemistry (IHC) was stopped early for lack of efficacy (results published 2014) [66]. Median survival was 7 months.

NABTC (North American Brain Tumor Consortium) 04-02 was a phase I/II study evaluating the combination erlotinib and temsirolimus for recurrent malignant gliomas [67]. Twenty-two patients were in the phase I study, while 47 patients were treated in the phase II portion. Patients were given erlotinib 150 mg daily (up to 200 mg daily as tolerated) and temsirolimus 15 mg weekly. There was significant dose-limiting toxicity in the form of rash and mucositis. Of 42 glioblastomas, 29 patients had stable disease; progression free survival at 6 months was 13%. Pre-treatment EGFRvIII, EGFR amplification, and phospho-EGFR did not correlate with survival.

None of the studies above evaluated post-treatment EGFR expression.

#### 2.3.3. Predicting Response of Erlotinib and Gefitinib in Glioblastoma

Extrachromosomal amplification and non-Mendelian distribution of extrachromosomal DNA (ecDNA) is present in both lung cancer and glioblastoma to similar degrees, both in terms of ecDNA counts and proportion of samples positive for ecDNA [53]. However, glioblastoma heterogeneity of EGFR is further manifested by the wide variability in expression EGFRvIII [68,69,70]. In addition, a patient-derived EGFRvIII-expressing xenograft mouse model demonstrated that erlotinib treatment at 150 mg/kg resulted in only temporary reduction of EGFRvIII expression—expression resumed pre-treatment levels once drug was stopped [71]. DNA copy number remained elevated throughout treatment, suggesting that tumor cell adaptation to the presence of the drug was by an epigenetic mechanism. These mechanisms of resistance, in concert with decreased drug concentration beyond the blood brain barrier, likely account at least in part for treatment failures.

Lassman et al. evaluated EGFR expression/signaling on gefitinib and erlotinib using Western blot (anti-pEGFR Tyr 1068 antibodies) on the high grade glioma datasets from NABTC 01-03 and 00-01 [72]. Erlotinib and gefitinib treatment did not consistently affect EGFR activity pre-/post- erlotinib or gefitinib, whether by pEGFR, pERK, or pAKT. This suggested that erlotinib and gefitinib did not effectively inhibit EGFR phosphorylation or signaling in these tumors.

Lack of effect on downstream signal transduction despite apparent appropriate inhibition appears to be mediated through alternative escape pathways, such as MET, IGF1R (insulin growth factor receptor 1), and PI3K. Mellinghoff et al. treated 49 recurrent glioblastoma patients with erlotinib (at doses from 150 to 500 mg) and gefitinib (from 150 to 1000 mg), and found that PTEN loss was associated with resistance to EGFR tyrosine kinase inhibitors [73]. There is also ample preclinical evidence that activation of the PI3K pathway [74], MET pathway [75,76], or the IGF1R pathway [77,78] confers resistance.

#### 2.3.4. Afatinib

Afatinib is a 2nd generation irreversible EGFR tyrosine kinase inhibitor [39]. Unlike erlotinib and gefitinib, afatinib has a relatively high affinity to wild-type EGFR, resulting in increased side effects such as diarrhea or skin rash [79].

Phase I/II study of afatinib versus TMZ versus the combination of afatinib plus TMZ in recurrent glioblastoma patients [80,81]. In phase I, patients were treated with afatinib at 20 mg daily (up to 50 mg daily). In phase II, 119 patients were split into three arms—afatinib 40 mg daily alone, combination afatinib 40 mg daily and TMZ 75 mg/m^2^ on a 21/28 day cycle, and TMZ 75 mg/m^2^ on a 21/28 day cycle alone. Progression free survival at 6 months was 3% (41 patients), 10% (39 patients), and 23% (39 patients), respectively. Median progression free survival was noted to be longer in afatinib-treated patients who had EGFRvIII positive tumors than EGFRvIII negative tumors. Post-treatment EGFR expression was not evaluated.

#### 2.3.5. Dacomitinib

Dacomitinib is a 2nd generation pan-ErbB tyrosine kinase inhibitor (EGFR, HER2, HER4). Its mechanism of action is similar to afatinib. Preclinical evidence in an orthotopic xenograft model of glioblastoma showed promise [82].

Dacomitinib was evaluated in a multicenter open-label phase II in 30 patients with recurrent glioblastoma (split across cohort A with EGFR gene amplification without EGFRvIII, and cohort B with both EGFR gene amplification and EGFRvIII), with the drug dosed at 45 mg daily. Median overall survival was 7.4 months overall (cohort A 7.8 months, cohort B 6.7 months). The authors concluded limited single-agent activity [83].

#### 2.3.6. Osimertinib

Osimertinib is a 3rd generation irreversible EGFR tyrosine kinase inhibitor, targeting the ATP-binding pocket of the kinase domain by formation of a covalent bond with Cys 797. Unlike afatinib, it has a much lower affinity for wild-type EGFR relative to its affinity the mutant T790M. The affinity for wild-type EGFR is over an order of magnitude less for osimertinib as it is for even gefitinib or erlotinib. This allows far higher tolerable doses for osimertinib. Additionally, osimertinib has been shown to have efficacy in brain metastases, conveying the ability to cross the blood brain barrier in clinically-relevant concentrations superior to that of other inhibitors [54,55,84]. There is interest in exploiting osimertinib’s excellent CNS (central nervous system) penetration for treatment of glioblastoma. However a potential concern is that osimertinib has relatively low affinity for the kinase domain of both wild-type EGFR and EGFRvIII (both of which have a wild-type kinase domain), as it was specifically designed to target the mutant T790M kinase domain (IC50 1 nM) and spare the wild-type kinase domain (IC50 184 nM) [43]. Its clinical tolerability and superior efficacy against T790M over second generation inhibitors such as afatinib is a direct consequence of this design. Despite this, preclinical studies report possible efficacy in glioblastoma [85,86].

No human trials have been completed to date. There are some reports in the literature of efficacy. One patient with EGFR A289V and EGFR C628F point mutations as well as EGFR copy number gain was treated with osimertinib at multifocal recurrence, with near complete response to osimertinib in one lesion but unfortunately discordant simultaneous progression at another site. The progressing lesion acquired an EGFRvIII mutation and continued to exhibit EGFR copy number gain [87].

#### 2.3.7. Rociletinib

Rociletinib is a 3rd generation inhibitor similar to osimertinib. Like osimertinib, it was developed as a highly-selective irreversible inhibitor of the ATP-binding pocket of the kinase domain by formation of a covalent bond with Cys 797 [88]. Unlike osimertinib, rociletinib is also an inhibitor of insulin growth factor receptor 1 (IGFR1). Blood brain barrier penetration is poor compared to osimertinib (brain/plasma ratio of 0.08, as compared to 3.41 for osimertinib) [55]. Development of rociletinib was discontinued in May 2016 after the Food and Drug Administration ruled against accelerated approval [44].

#### 2.3.8. Lapatinib

Lapatinib is a reversible dual EGFR/HER2 inhibitor. Like the small molecule inhibitors detailed above, lapatinib targets the kinase domains [89,90]. However, distinct from many of the inhibitors above that target the active kinase conformation (αC-helix in), lapatinib targets the inactive kinase conformation (αC-helix out). Targeting the inactive kinase conformation has limited utility in the context of NSCLC (associated with mutant active kinase conformations), but has some rationale for use in gliomas and specifically in combination with ABT-414 (see below). A small phase I/II study of lapatinib dosed between 1000 mg twice daily and 1500 mg twice daily in 17 patients with recurrent glioblastoma identified no significant activity (four patients with stable disease, with the remaining vast majority of patients progressing) [91]. Another phase I study with similar dosing of 16 recurrent glioblastomas on a phase I study noted median OS (overall survival) of 5.9 months and median PFS (progression free survival) of 2.4 months [92]. A pilot phase II study of pulsed-dose lapatinib in 12 newly-diagnosed glioblastomas, dosed at 2500 mg twice daily for two consecutive days per week through concurrent and adjuvant standard of care therapy, determined that the combination was reasonably well-tolerated [93].

#### 2.3.9. Neratinib

Neratinib is an irreversible EGFR, HER2, and HER4 inhibitor that forms a covalent adduct with the conserved cysteine corresponding to the 797 position on the EGFR kinase [94]. Similar to lapatinib, neratinib targets the inactive kinase conformation (αC-helix out). Neratinib is one of the investigational agents on the ongoing INSIGhT (Individualized Screening Trial of Innovative Glioblastoma Therapy) clinical study [95].

#### 2.3.10. Cetuximab

Cetuximab is a monoclonal antibody targeting domain III, interfering with ligand binding. Similar to findings with TKIs, preclinical studies of cetuximab noted failure to alter EGFR downstream signal transduction in the context of gliomas, with reduction in cell viability achieved after inhibiting PI3K and MET pathways [96]. Nevertheless, clinical cases of response to single agent cetuximab responses in EGFR-expressing recurrent glioblastoma have been reported [56].

A phase II study of cetuximab in combination with bevacizumab and irinotecan for recurrent glioblastomas found no improvement in outcomes compared to bevacizumab and irinotecan alone [97].

#### 2.3.11. Rindopepimut

Rindopepimut (also known as CDX-110), a cancer vaccine consisting of an EGFRvIII-specific peptide conjugated to KLH (keyhole limpet hemocyanin) enrolled 745 patients with newly-diagnosed glioblastoma between 2012 and 2014 in an international phase III study (ACT IV). There was no improvement in survival between the treatment and the control groups (HR 1.01, *p* = 0.93). Furthermore, in a comparison of post-treatment tumor samples, EGFRvIII expression was undetectable at similar levels between treated and control (57% and 59%, respectively). The anti-EGFRvIII titer also did not differ between the two groups [98].

Although only a small percentage of the trial population had post-treatment tumor analysis (about 10%), this trial was one of the few to attempt confirmation of effects on EGFR. It is unclear whether the failure of rindopepimut is related to its inability to induce an effective immune response.

#### 2.3.12. Depatuxizumab Mafodotin (ABT-414)

Depatuxizumab mafodotin (Depatux-M, or ABT-414), is an ADC (antibody drug conjugate) consisting of the monoclonal antibody 806 (targeting EGFR-overexpressing cells) conjugated to MMAF (monomethyl auristatin F) [99,100]. The epitope of the targeting monoclonal antibody ABT-806 is a cryptic region on extracellular domain II, near the domain II–domain III interface [101]. Normally buried in both the open untethered and the closed tethered conformations of the extracellular domain, this epitope sits blocked by domain I and the N-terminal part of domain II (residues 6–273, a region referred to as N-TR1). Deletion along N-TR1 (as occurs in EGFRvIII deletion mutation) and twisting/bending of N-TR1 (as occurs in mutations in domain I or domain II) result in exposure of this cryptic epitope—compellingly demonstrated by Orellana and colleagues in a mechanistic manner [27], and corroborating earlier observations noting a predisposition of 806 for glioblastoma cell lines both with and without EGFRvIII deletion [100,102]. Proliferation of tumor lines were not exhibited by 806 in vitro—cytotoxicity of ABT-414 depends on delivery of the toxic payload. In an ectopic EGFRvIII U87MG cell line, the IC50 (half maximal inhibitory concentration) of ABT-414 cytotoxicity was 0.3 nM—compared to 222 nM in the standard U87MG cell line with wild-type EGFR [99]. As antibodies are too large to cross the blood brain barrier effectively, this is a concern for ADCs such as Depatux-M. Evidence of blood brain barrier penetration for Depatux-M is primarily by single-photon emission computed tomography (SPECT) imaging with intravenously-administered ^111^indium-labeled ABT-806—with uptake seen in both an orthotopic xenograft mouse model [100] and in the human brain [57]. To our knowledge the concentration of Depatux-M in CSF (cerebrospinal fluid) or brain has not been directly measured. There is evidence however that ABT-414 reaches its target—EGFR amplification from surgical samples post-treatment in Depatux-M patients is lower (44%) than that of non-Depatux-M patients (87%) [103].

A multicenter phase 1 open-label clinical trial in 38 patients with recurrent glioblastoma noted a progression free survival at 6 months of 30.8%, in patients treated either with Depatux-M alone or in combination with temozolomide—median overall survival for all recurrent glioblastomas was 10.7 months (17.9 months in the combination arm, and 7.2 months in the monotherapy arm) [104]. PFS was 3.7 months in the combination arm, 2.3 months in the monotherapy arm, and 2.3 months in all comers. Four of the five patients with a >50% reduction in tumor volume from baseline were EGFR amplified. Ocular toxicity was observed in over 90% of patients, though largely reversible with holding of treatment. Although pre-treatment EGFR status was assessed and serum pharmacokinetic studies were done, CSF and brain concentration of ABT-414 was not assessed. A parallel phase I/II study in 38 Japanese patients with recurrent glioblastoma (INTELLANCE-J) found that the drug was well-tolerated with median progression free survival of 4 months, median overall survival of 15.5 months—and an overall survival of 93%, 62.5%, and 28% at 6, 12, and 24 months, respectively [105].

The randomized phase II study (INTELLANCE-2) studied Depatux-M in 260 patients with centrally-confirmed EGFR-amplified recurrent glioblastoma [106]. Patients were randomized 1:1 to either Depatux-M in combination with temozolomide, Depatux-M alone, or a control arm (either lomustine or temozolomide). The combination Depatux-M with temozolomide arm showed a statistically-significant improvement in overall survival at 2 years compared with the control arm (19.8% versus 5.2%, respectively)—suggesting a role for the combination—however, no evidence of efficacy was noted in the monotherapy arm [107].

INTELLANCE-1, the phase III randomized, double-blind, placebo-controlled study of Depatux-M in combination with radiotherapy/temozolomide versus radiotherapy/temozolomide alone in EGFR-amplified newly-diagnosed glioblastomas unfortunately found no survival benefit at an interim analysis; the study was stopped for futility [108].

There is evidence that the intermediate conformation ectodomain (targeted by ABT-414) is allosterically linked to the inactive kinase conformation—Orellana demonstrated that stabilizing the inactive kinase conformation with lapatinib correlated convincingly with increased binding of ABT-806 (which targets the intermediate conformation on the ectodomain). Stabilizing the active kinase conformation with gefitinib inhibited ABT-806 binding [27]. There is therefore rationale for ABT-414 combination with tyrosine kinase inhibitors against the inactive kinase conformation (such as lapatinib or neratinib).

## 3. Discussion

Targeting of EGFR in the non-small cell lung cancer space has made incredible strides, with clear correlation between drug action, resistance mutations, and clinical activity. The mechanistic and therapeutic insights in the kinase domain have not yet translated into clear therapeutic efficacy in glioblastoma, where it is still unclear how the specific EGFR ectodomain alterations—EGFRvIII in particular—can be properly targeted. Despite being found in over half of all glioblastomas [109,110], EGFR alterations remain an elusive target. Challenges to targeting of EGFR include heterogeneity of EGFR mutant and wild-type expression [58,59,111], adverse effects arising from collateral inhibition on wild-type EGFR in normal tissues, reduced penetration of monoclonal antibodies across the blood brain barrier (particularly at the infiltrating edges of the tumor where blood brain barrier is intact), and escape mechanisms such as the PI3K and MET pathways.

At first glance, it is surprising that targeting of EGFR has lagged in glioblastoma. The prevalence of EGFR mutation is far higher in glioblastoma than it is for NSCLC—suggesting its greater importance in glioblastoma biology. Furthermore, heterogeneity, at least by some measures, is at least as much a factor in NSCLC as it is in glioblastoma, if not more—for example, tumor mutational burden is actually higher for NSCLC than it is for glioblastoma [112]. A higher tumor mutational burden suggests an increased pool of mutations from which to draw upon in response to external pressures placed by targeted therapy. The resilience of EGFR amplification and EGFRvIII in glioblastoma may represent a failure of our drugs to act against the specific EGFR alterations in glioblastoma, rather than an indictment of the validity of targeting the EGFR axis in glioblastoma. While many other factors are likely at play (such as redundancy of regulatory circuits [113,114,115,116]), it is clear that present-day inhibitors simply do not inhibit downstream signal transduction of EGFR in glioblastoma effectively.

As kinase domain mutations are only rarely seen in glioblastoma, it is unlikely that small molecule inhibitors targeting the kinase domain—whether reversibly like the 1st generation TKIs or irreversibly like the 2nd generation TKIs—will see much success. The lack of specificity for a glioblastoma-specific target results in dose-limiting toxicity.

The dimerization interface of EGFRvIII, whether in the context of a homodimer with a partner EGFRvIII or as a heterodimer with wild-type (over-expressed) EGFR, is a possible site of a tumor-specific epitope. It is worth noting that EGFRvIII’s heterodimerization and ligand-independent activation of ErbB partners has many parallels with a naturally-occurring member of the ErbB family that is overexpressed in cancer—HER2. The monoclonal antibody pertuzumab, used in HER2 overexpressed breast cancer, targets the extracellular domain II on HER2 and interferes with its ability to heterodimerize with other members of the ErbB family by occluding the dimerization arm [117]. This precedence lends credence to this analogous strategy against the EGFRvIII-EGFR dimerization interface.

The ligand-independent EGFR oligomer, which dominates at high EGFR concentrations (relevant in EGFR overexpressed glioblastoma), is likely to have epitopes distinct from those of classic ligand-dependent dimers. Ligand-independent EGFR oligomers arrange in a manner distinct from the classic “back-to-back” configuration of ligand-dependent dimers seen in cells with lower EGFR expression [118]. As ligand-independent oligomers would dominate in glioblastoma cells where EGFR concentrations are high, but not in normal cells that rely on classic ligand-dependent dimerization, the EGFR oligomer may be another source of specific epitopes not seen on normal cells.

Identification and development of small molecule inhibitors for the above interfaces will be challenging—not only is a greater understanding of the involved structures still needed (including that of post-translational modifications that might be relevant on the extracellular domain, such as glycosylation), but protein–protein interfaces remain a challenging frontier in small molecule drug development [119]. Developing small molecule inhibitors for the shallow interfaces of the extracellular domain is a difficult task compared to doing the same for the “deep” binding pocket of the tyrosine kinase active site. All existing tyrosine kinase inhibitors target the ATP-binding pocket of the kinase domain for this reason. Monoclonal antibodies can and have been developed against epitopes on the extracellular domains (i.e., cetuximab against domain III, and ABT-806 against domain II), but antibodies are large and must contend with the blood brain barrier [120].

Biologic endpoints are crucial to clarify reasons for failure. Ideally, all future clinical studies targeting EGFR should include pre-/post- tissue EGFR expression, and measurement of intra-tumoral drug concentration in both enhancing and non-enhancing regions of the tumor on the MRI (magnetic resonance imaging). The pre-treatment and post-treatment status of important known escape pathways (such as PI3K and MET) would optimally be assessed as well.

Efforts to target EGFR in glioblastoma continue to be investigated. Clinical trials currently active and in development for EGFR in glioblastoma include the study of several novel agents, including the TKI epitinib, bi-armed activated T cells against EGFR/CD3, EGFR(V)-EDV-Dox, the monoclonal antibody GC1118, doxorubicin-loaded anti-EGFR immunoliposomes, and the anti-EGFRvIII/CD3 bispecific T-cell engager (BiTE) AMG 596, among others [121]. The results of these studies, and future steps to deconstruct the mechanistic underpinnings of EGFR regulation in glioblastoma will hopefully lead to improved understanding and traction in stepwise rational targeting of this important pathway, with better outcomes in this highly vulnerable patient population.

## Figures and Tables

**Figure 1 ijms-21-08471-f001:**
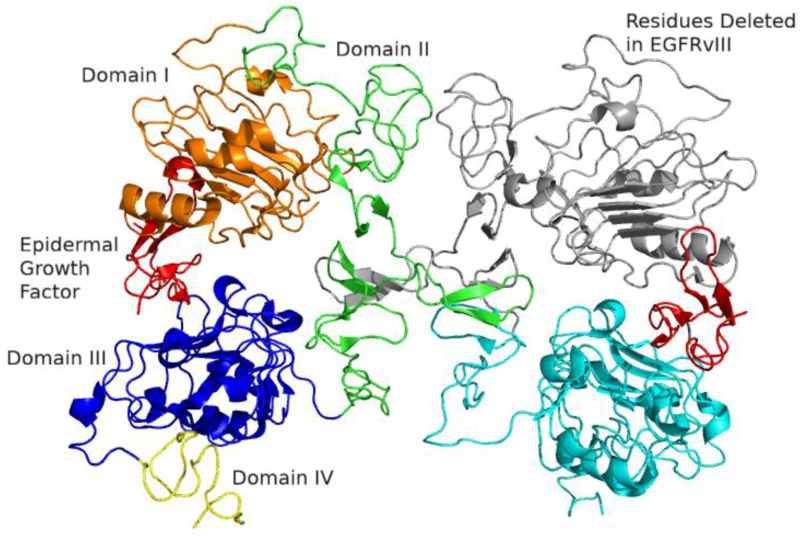
Extracellular domains of epidermal growth factor receptor (EGFR) homodimer, in 2:2 EGFR:EGF (epidermal growth factor ) ligand complex. The receptor on the left is colored as domain I (orange), II (green), III (blue), and IV (yellow) respectively, with its partner wild-type EGFR monomer on the right in gray and teal. Amino acids 6–273, deleted in EGFRvIII, are highlighted in gray on the right. The extracellular domain consists of four domains I through IV—or L1, S1, L2, and S2—with S1 (CR1) and S2 (CR2) being Cys-rich [1,2]. The ligands, epidermal growth factor, are colored in red. Ligands bind to domains I and III (L1 and L2), away from the site of dimerization at domain II, as demonstrated with crystal structure by Garrett [3] and Ogiso [1]. This mode of ligand binding in EGFR, distant from the dimerization site, is distinct among receptor tyrosine kinases [4]. As can be seen in the figure, ligand-dependent dimerization arranges the dimer with the ligands in a so-called “back-to-back” arrangement, with the EGF ligands pointing away from each other [1,3]. Figure generated from PDB ID (Protein Database Identifier) 1IVO [1] with PyMOL [5].

**Figure 2 ijms-21-08471-f002:**
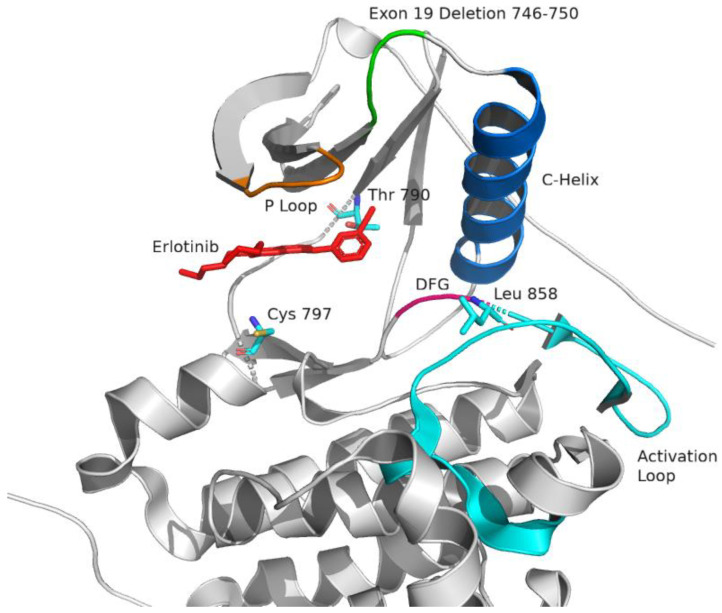
Kinase domain of EGFR, in the active configuration. The αC (C-helix) is in the inward activated position, in complex with erlotinib. Phosphate-binding P-loop (orange), C-helix (blue), and the activation loop (teal) are highlighted. DFG is in the inward activated position and colored in magenta. Erlotinib is colored red. Several clinically-relevant sites of mutation are highlighted. Exon 19 deletions typically affect amino acids 746–750, highlighted in green. Leucine 858, threonine 790, and cysteine 797 are highlighted and shown as stick figures. Figure generated from PDB ID 1M17 [8] in PyMOL [5].

**Figure 3 ijms-21-08471-f003:**
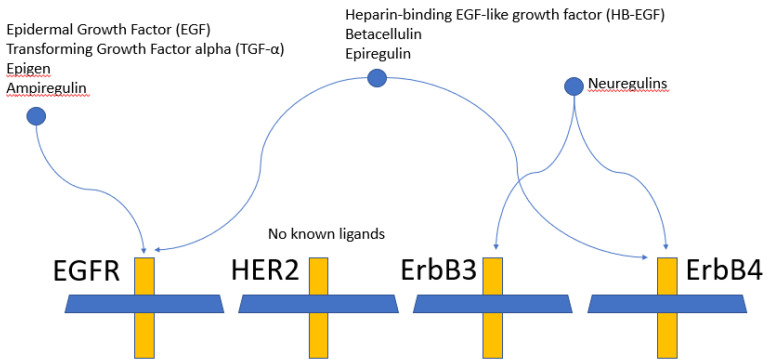
Ligands for the ErbB receptor family. EGFR (ErbB1/HER1) has 7 known ligands. ErbB2 (HER2) uniquely has no known ligands. ErbB3 (HER3) uniquely has no kinase domain activity.

**Figure 4 ijms-21-08471-f004:**
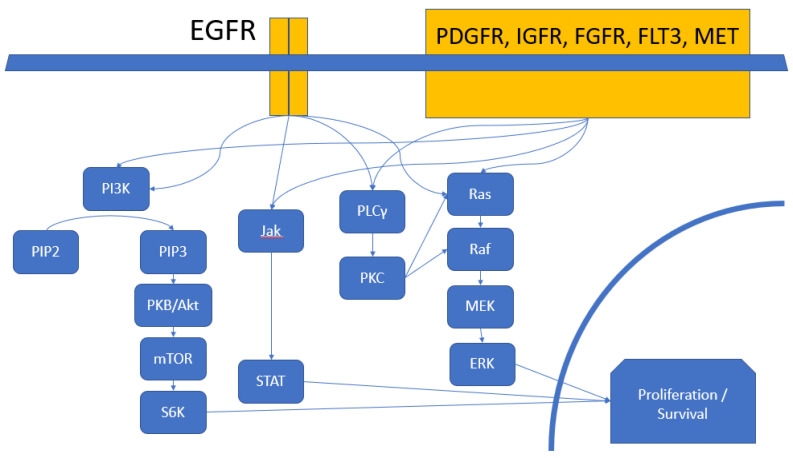
Downstream signaling of EGFR and other receptor tyrosine kinases [9,10,11]. Signal transduction into the nuclear compartment eventually leads to genetic and epigenetic changes leading to tumor proliferation, genetic immortality, invasion, angiogenesis, and avoidance of immune surveillance. Pathway crosstalk and redundant signaling offers escape pathways mediating treatment resistance.

**Table 1 ijms-21-08471-t001:** EGFR drugs and mechanisms of action.

Name		Mechanism of Action
Gefitinib	Type I TKI	Targets EGFR. Reversible ATP-site competitive inhibitor of EGFR kinase domain.
Erlotinib	Type 1 and Type I ½B TKI	Targets EGFR. Reversible ATP-site competitive inhibitor of EGFR kinase domain.
Afatinib	Type VI (irreversible) TKI	Targets EGFR. Irreversible ATP-site EGFR kinase inhibitor. Covalent bond with Cys 797 on EGFR kinase domain.
Rociletinib	Type VI (irreversible) TKI	Targets EGFR. Irreversible ATP-site EGFR kinase inhibitor. Covalent bond with Cys 797 on EGFR kinase domain.
Osimertinib	Type VI (irreversible) TKI	Targets EGFR. Designed for low affinity for wild-type EGFR compared to L858R/T790M, with improved therapeutic index. Irreversible ATP-site EGFR kinase inhibitor. Covalent bond with Cys 797 on EGFR kinase domain.
Lapatinib	Type I ½ TKI	Targets EGFR/HER2/HER4. Reversible ATP-site competitive inhibitor, targeting the inactive kinase conformation (αC-helix out). Dual EGFR/HER2 inhibitor.
Dacomitinib	Type VI (irreversible) TKI	Pan-ErbB TKI. Irreversible pan-ErbB tyrosine kinase ATP-site kinase inhibitor. Covalent bond with Cys 797 on EGFR kinase domain.
Neratinib	Type VI (irreversible) TKI	Targets EGFR/HER2/HER4. Irreversible pan-ErbB tyrosine kinase ATP-site kinase inhibitor. Covalent bond with Cys 797 on EGFR kinase domain.
Cetuximab	Monoclonal Antibody	Targets extracellular domain III, interfering with ligand binding.
Rindopepimut (CDX-110)	Peptide Vaccine	Peptide vaccine of EGFRvIII-specific peptide conjugated to KLH
Depatuxizumab mafodotin (ABT-414)	Antibody Drug Conjugate	Targets cryptic region on extracellular domain II exposed by EGFRvIII mutation, extracellular domain I mutations, and domain II mutations. Conjugated to MMAF (monomethyl auristatin F) payload.

Table 1. Brief summary of agents against EGFR. Small molecule kinase inhibitors follow Roskoski classification [18]. Type I binds in ATP-binding pocket in its active conformation (DFG-Asp “in,” αC-helix “in,” linear R-spine). Type I ½ binds in ATP-binding pocket in inactive conformation with DFG-Asp in the “in” conformation but αC-helix “out” or R-spine distorted; subdivided into A if drug extends into back cleft, and B if not. Type II binds in ATP-binding pocket in inactive conformation with DFG-Asp in the “out” conformation; subdivided into A if drug extends into back cleft, and B if not. Type III is an allosteric inhibitor acting next to ATP-site, and Type IV is an allosteric inhibitor not acting next to ATP-site. Type V is bivalent inhibitor spanning two regions. Type VI is an irreversible inhibitor.

**Table 2 ijms-21-08471-t002:** Summary of select EGFR clinical trials in glioblastoma (GBM).

Trial	Phase	Therapeutic Approach	Response	Status
Mayo/NCCTG N0074 [52]	II	Gefitinib 500–1000 mg daily in new GBM	PFS12 16.7% OS12 54.2%	Closed
N0177 [53]	I/II	Erlotinib 150 mg daily in new GBM	OS12 61%	Closed
Reardon et al., 2014 [54] and Eisenstat et al., 2011 [55]	II	Afatinib 20–50 mg daily alone, in combination with TMZ, or TMZ alone in recurrent GBM	PFS6 3%, 10%, 23% respectively	Closed
Karavasilis et al., 2013 [56]	I	Lapatinib 1000–1500 mg twice daily in recurrent GBM	mOS 5.9 months mPFS 2.4 months	Closed
ACT IV [57]	III	Rindopepimut in new GBM	No improvement compared to control	Closed
INTELLANCE-2 [58]	II	Depatux-M alone, in combination with TMZ, or control (TMZ or lomustine), in recurrent GBM	OS24 19.8% with combination Depatux-M and TMZ compared to control OS24 5.2%. No benefit in monotherapy Depatux-M arm.	Closed
INTELLANCE-1 [59]	III	Depatux-M in combination with RT/TMZ vs. RT/TMZ alone, in new GBM	Stopped for futility; no survival benefit at prespecified endpoint	Closed
INSIGhT	II	Neratinib in new GBM	Pending	Ongoing

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
