# Peer review of "Mechanisms of EGFR Resistance in Glioblastoma"

_ijms, 2020, doi:10.3390/ijms21228471_

Round 1
Reviewer 1 Report
The manuscript is well organized and devolopped.
This s not the classic review were the authors listed all the preclinical and clinical studies but this manuscript had a broader approach. The authors begin from a more biochemical area describing the EGFR structure, ligands and the EGFR pathways to the clinical part deepening the most relevant EGFR target agents and the corresponding clinical studies. For this reason, I appreciate the originality of this manuscript.
Author Response
I greatly appreciate these comments. The reason for the disparity in targetability of EGFR in glioblastoma as opposed to that of other tumor types remains an open question. We hope that a structural understanding of EGFR can highlight some of the challenges that will need to be overcome in effective treatment.
Reviewer 2 Report
This systematic review by Peter C. Pan and Rajiv S. Magge. is a comprehensive and state-of-the-art presentation of the new insight into mechanisms of EGFR resistance in glioblastoma from a pre-clinical approach. Their approach entails a new and substantial contribution to current literature on the subject matter; however, I have the following major but serious concerns:
- This review study is largely confirmatory of a previously published study by 2019 Apr 12;8(4):350.; Wiley Interdiscip Rev Syst Biol Med. 2018 Jan;10(1):10.1002/wsbm.1398.;3(5):349-58.; Mol Oncol. 2018 Jan;12(1):3-20., Preclinical studies underscore the importance of EGFR overexpression has been shown to be associated with enhanced metastatic potential in glioblastoma, with EGFR at the top of a downstream signaling cascade that controls basic functional properties of glioblastoma cells and therefore lacks significant novelty of this review.
- This review is too long and, in many places repetitive. Authors should review the paper content for redundancy and ensure only essentials are left. Please reduce the paper about 15-20 pages.
- Though there is not standard limit for references in review articles, i feel over 125 references is just too many. Authors should look at cutting this down to < 100.
- Please focus and discuss some paragraphs of novel treatment approach of Glioblastoma by targeting EGFR resistance.
- Please also discuss the targeting EGFR resistance with plant-derived natural products regarding the emerging trends in Glioblastoma cancer therapy
- There are also a few errors in English language grammar that require the authors' attention.
- All abbreviations must be defined when they are first used.
- Authors may want to provide a more representative Graphical Abstract
This manuscript cannot be accepted in its present form, I recommend REJECTION.
Author Response
We greatly appreciate the reviewer's time in preparing a thorough and thoughtful critique of our review. We have addressed each concern in kind as follows:
1. The reviewer linked to excellent review papers (Liu and Mischel 2018, and Sigismund et al 2017; unfortunately we were not able to locate the first paper mentioned). We agree with the reviewer that EGFR plays a large role in glioblastoma. However our goal is to delineate the structural reasons why current treatments have not been successful.
2. We appreciate the reviewer's comments. We've gone through and eliminated extraneous language and redundant prose.
3. We appreciate reviewer's commments. However we'd like to do our due diligence and give credit to the many papers that have contributed to the field. Please let us know if this continues to be a concern and we'd be happy to address specific references.
4. Appreciate the reviewer's comments, on pages 15 and 16 we've indicated novel potential ways of treating EGFR resistance in GBM.
5. We understand there are several alternative treatments that are being considered for glioblastoma, however we are not aware of any that have robust scientific support for targeting of EGFR in the context of GBM. If the reviewer has any specific studies in mind, we'd be happy to include them.
6. We reviewed the paper again extensively and have made grammatical changes. If there are specific areas that the reviewer would like further addressed, we would be happy to make the appropriate changes.
7. We reviewed through the paper and ensured that all abbreviations were defined at first use.
8. We appreciate the reviewer's comments. As this is a review paper, it is difficult to condense into a single graphical abstract. However if there are specific figures that the reviewer would like us to include, we would be happy to consider it.
Reviewer 3 Report
Overall this is a very well written review covering multiple aspects of EGFR biology and at the same time discussing in greater detail the hurdles involved in the therapeutic targeting of EGFR in glioblastoma.
As the authors have given a detailed account of multiple drugs/Abs developed for targeting the EGFR pathway, It would be a lot easier for readers to get an overview if the information given here could also be presented in the form of a table including details like- therapeutic approach, target, phase of trail, response, current status etc.
Further something very minor in line 74 which says- Epidermal growth factor ligands colored red. (please check this sentence as it seems to make no sense to be mentioned in that paragraph.
Author Response
We greatly appreciate the reviewer's thoughtful comments.
1) We have included a table highlighting the major studies targeting EGFR in glioblastoma, including details such as phase, response, status, and a link with reference to the results.
2) We appreciate this finding. The sentence was originally to be part of the nearby figure caption but was misplaced. We have corrected this error.